# FSL-1 Pre-Administration Protects Radiation-Induced Hematopoietic Organs Through the Modulation of the TLR Signaling Pathway

**DOI:** 10.3390/ijms26115303

**Published:** 2025-05-31

**Authors:** Venkateshwara Rao Dronamraju, Gregory P. Holmes-Hampton, Emily Gu, Vidya P. Kumar, Sanchita P. Ghosh

**Affiliations:** Armed Forces Radiobiology Research Institute, Uniformed Services University of the Health Sciences, Bethesda, MD 20889, USA; venkateshwara.dronamraju.ctr@usuhs.edu (V.R.D.); gregory.holmes-hampton@usuhs.edu (G.P.H.-H.); emily.gu.ctr@usuhs.edu (E.G.); vidya.kumar.ctr@usuhs.edu (V.P.K.)

**Keywords:** total body irradiation, murine, mixed-field irradiation, gamma irradiation, neutron irradiation, fibroblast-stimulating lipopeptide 1, toll-like receptor agonist

## Abstract

Substantial progress has been made in the development of radiation countermeasures, resulting in the recent approval of several mitigators; however, there has yet to be an approved prophylactic radioprotectant. Research on countermeasure performance in mixed neutron and gamma radiation fields has also been scarce. Fibroblast-stimulating lipopeptide (FSL-1) is a novel synthetic agonist for toll-like receptor 2/6. In previous studies, the administration of FSL-1 before and after gamma radiation significantly improved survival outcomes for mice through the activation of the NF-κB pathway. In the current study, we tested FSL-1’s radioprotective abilities in a mixed radiation field that models one produced by a nuclear detonation in 11–14-week-old C57BL/6 male and female mice. We demonstrate that a single dose of 1.5 mg/kg of FSL-1 administered 12 h prior to 65% neutron 35% gamma mixed-field (MF) irradiation enhances survival, accelerates recovery of hematopoietic cell and stem cell populations, reduces inflammation, and protects innate immune function in mice. FSL-1’s ability to recover blood and protect immune functions is important in countering the high rate of incidence of sepsis caused by MF radiation’s damaging effects. These results demonstrate that FSL-1 is a promising prophylactic countermeasure where exposure to MF radiation is anticipated.

## 1. Introduction

Nuclear facility accidents, terrorist use of improvised nuclear devices (INDs), and military nuclear detonations are scenarios that can result in exposure to a mixed field (MF) of neutron and gamma radiation. Combined neutron and gamma radiation is more damaging than gamma radiation alone due to the high linear energy transfer (LET) radiation of emitted neutrons [1]. In view of the escalating threat of nuclear weapon use in major conflict zones around the world, there is an unmet need for the development of prophylactic radiation countermeasures for the protection of military and first responders who would be leading rescue and evacuation efforts. Recently, the Food and Drug Administration (FDA) has approved several countermeasures and biosimilars [2,3,4,5] against hematopoietic acute radiation syndrome (H-ARS) that can be administered as mitigators post-radiation exposure; however, there are no countermeasures that have been approved for prophylactic use. Therefore, the development of prophylactic countermeasures is an urgent national need to protect our warfighters before sending them in harm’s way.

Neutron radiation produces secondary neutrons and spurs of hydroxyl radicals, which can lead to double-stranded breaks (DSBs) in DNA and other combinations of base damage known as locally multiply damaged sites (LMDSs) [6]. DNA base sequences can be lost at DSBs and LMDSs, and damage may be irreversible [7]. In addition, neutron radiation changes DNA flexibility and is 25 times more effective than gamma radiation in causing DNA breaks via uncoiling of the helix [8,9]. DNA strand breaks caused by neutrons also result in greater G1 cell cycle arrests [10]. Overall, the damage to organs due to MF exposure may be irreversible based on the extent of injury without medical intervention [11].

There are few promising prophylactic countermeasures that have been developed in pre-clinical animal models of total body gamma radiation. Since nuclear detonations will result in a mixed gamma and neutron radiation field, it is vital to understand the ability of radiation countermeasures to function in these harsher conditions. Cary et al. demonstrated that medical countermeasures for ARS should be tested against various radiation qualities (both high and low LET), which can be used for specific radiological nuclear scenarios. The authors showed that while granulocyte colony-stimulating factor (G-CSF) and thrombopoietin mimetic (TPOm) ALXN4100TPO are both effective mitigators against gamma radiation, only G-CSF effectively protected against MF radiation in mice exposed to total body radiation (TBI) [12]. Thus, the radioprotectant that can be used for a nuclear detonation response must have proven efficacy against exposure to MF radiation.

MF radiation causes significantly more hematopoietic injury and increases rates of infection and sepsis [13,14]. In search of a radiation prophylactic agent that can be effective for both gamma and MF exposure, we tested a toll-like receptor (TLR) agonist fibroblast-stimulating lipopeptide (FSL-1) in mice using both radiation sources. FSL-1 is a synthetic diacylated lipoprotein (S,R)-(2,3- bispalmitoyloxypropyl)-Cys-Gly-Asp-Pro-Lys-His-Pro-Lys-Ser-Phe originally derived from mycoplasma salivarium [15,16]. TLRs are a family of dimerizing receptors expressed on immune cells and have been shown to be promising candidates for stimulating innate immunity. They recognize pathogen-associated molecular patterns [17] and subsequently release various immunomodulators and inflammatory cytokines [18]. Entolimod (Cleveland Biolabs Protectan CBLB502), a salmonella flagellin derivative, triggers radioprotection by activating TLR5 [19]. While TLR5 recognizes flagellin [20], TLR1, TLR2, TLR4, and TLR6 recognize lipid species [21]. CBLB613, a natural lipopeptide of mycoplasma arginini, has shown promise as a radiation countermeasure via heterodimer TLR2/6 signaling [22]. TLR2 is highly expressed in immune cells. Its activation triggers the nuclear factor-kappa B (NF-κB), which upregulates anti-apoptotic genes [23], anti-oxidants [24], cytokines [25,26], and growth factors [27]. Synthetic lipopeptides have been synthesized (sLP) to activate TLR2 as adjuvants in vaccines to stimulate immune response [28]. sLPs have been shown to be radioprotective in mice by activating TLR2 [22]. In addition, Cleveland Biolabs Protectan (CBLB612) was developed as an agonist of TLR2 [22] and has recently entered a Phase II trial in Russia to prevent myelosuppression in patients undergoing chemotherapy.

Heterodimer TLR2/6 recognizes the diacylated cysteine residue on FSL-1, consequently activating the NF-κB pathway to promote inflammation and cell proliferation. Animal studies in mice and non-human primate models showed hematopoietic recovery when FSL-1 was given 24 h post-radiation exposure [29,30]. In a recent publication, the authors demonstrated that FSL-1 administered 24 h after gamma radiation effectively increased survival rates in both males and mice, accelerated hematopoiesis in the bone marrow and the spleen, and protected the hematopoietic system [29]. In our previous study, we have shown that FSL-1 administered 24 h prior to gamma radiation significantly increased survival and enhanced hematopoietic recovery in male and female C57BL/6 mice. We determined the optimal dose at this time of administration to be 1.5 mg/kg body weight through a dose–response study [31]. In the present study, we demonstrate FSL-1’s prophylactic radioprotective efficacy using a more damaging MF radiation field of 65% neutron and 35% gamma produced by an in-house Training, Research, Isotopes, and General Atomics (TRIGA) nuclear reactor, which can simulate fields with an energy distribution similar to nuclear detonations [32]. In addition, this data shows that the immune stimulatory effects of FSL–1 protected the hematopoietic system and increased survival by accelerating hematopoietic recovery following radiation exposure. Finally, these data support the development of FSL-1 as a promising medical countermeasure that can be developed for warfighters and first responders and can be given before sending them for rescue or recovery operations in an MF radiation-contaminated field.

## 2. Results

### 2.1. FSL-1 Provides Significant Survival Benefit When Given 6 or 12 h Pre-TBI in C57BL/6 Mice Exposed to Either Gamma or MF Radiation

To evaluate the effect of FSL-1 administered as a single dose at 6 or 12 h pre-total body gamma radiation, male and female C57BL/6 mice aged 11–13 weeks were pre-treated with a 0.1 mL single SC dose of FSL-1 (1.5 mg/kg) or its vehicle, sterile injectable saline, and were exposed to lethal radiation doses (9.0 Gy for male and 8.3 Gy for female mice) at a dose rate of ~0.6 Gy/min in the AFRRI high-dose rate cobalt-60 facility (Figure 1A,B). Male mice administered FSL-1 at 12 h or 6 h before irradiation (Figure 1A) showed significantly higher survival at 30 days post-TBI (92% and 72%, respectively) compared to age-matched saline cohorts (0% survival). Significance measured as a *p*-value was <0.0001 and 0.0001 for log-rank (Mantel–Cox) and Fisher’s exact tests, respectively. Similarly, female mice exposed to Co-60 gamma radiation in the same facility showed higher survival when pre-treated with FSL-1 (1.5 mg/kg) 12 h pre-TBI (Figure 1B). Significance measured as a *p*-value was 0.0042 and 0.005 for log-rank (Mantel–Cox) and Fisher’s exact tests, respectively. To determine the efficacy of FSL-1 in a mixed-field gamma and neutron environment, male and female C57BL/6 mice were exposed to 5.5 Gy mixed-field radiation (65% neutron and 35% gamma) in the TRIGA reactor facility (Figure 1C,D). FSL-1 administered as a single S.C. dose 12 h pre-TBI provided significant survival benefit in male mice treated with FSL-1 at 5.5 Gy mixed-field radiation and showed 71% survival compared to 0% survival in the age-matched saline-treated cohort. Statistical significance measured using the log-rank (Mantel–Cox) test was found to be *p* = 0.0043, and for Fisher’s exact test, it was *p* = 0.0029. Similarly, in female mice, FSL-1 treatment 12 h pre-TBI resulted in 75% survival compared to 29% survival for the age-matched saline cohorts. Statistical significance measured using the log-rank (Mantel–Cox) test was *p* = 0.0021, and Fisher’s exact test was *p* = 0.0034. To evaluate the sternal bone marrow recovery following exposure to a lethal dose of radiation, H&E-stained slides from 60-day survivors (male mice) were quantified and showed a significant increase (*p* = 0.009) in megakaryocyte counts (119.6 ± 44.49) in animals treated 12 h pre-TBI compared to animals treated 6 h pre-TBI (72.50 ± 23.90) (Figure 1E). H&E staining of sternal bone marrow in female mice survivors 60 days post-TBI showed significantly higher (*p* = 0.003) megakaryocyte counts in animals treated with FSL-1 pre-TBI (99.83 ± 17.43) compared against animals administered saline (63.17 ± 15.37) (Figure 1E). FSL-1 treatment 12 h pre-TBI also significantly increased the number of CFUs compared to the saline-treated cohorts in female mice 60 days post-TBI (*p* = 0.014, Figure 1G). However, there was no significant difference between 12 h pre-TBI and 6 h pre-TBI treatment in male mice (*p* = 0.844) (Figure 1F).

### 2.2. Accelerated Recovery from Radiation-Induced Pancytopenia, Hematopoietic Progenitor Cell Loss, and Sternal Megakaryocyte Depletion

To understand the effect of FSL-1 on blood cell recovery, male mice were administered a single SC dose of FSL-1 (1.5 mg/kg) or saline 12 h prior to exposure at a sublethal dose of 4.8 Gy using a mixed field (65% neutron and 35% gamma). As we have demonstrated previously, sublethal doses of radiation results in higher survival while still producing injury in the animals [33]. Whole blood as well as sternal and femoral bone marrow was collected from animals at 1, 4, 7, 14, and 30 days post-radiation (*n* = 5 per timepoint). Figure 2A–D show the hematology parameters measured in the whole blood of these animals. Male mice treated with FSL-1 12 h pre-TBI showed accelerated recovery of WBCs, NEUs, LYMs, and PLTs (Figure 2A–D).

WBCs: Both irradiated groups given either FSL-1 or saline showed a progressive decrease in WBC counts. However, mice treated with FSL-1 showed higher WBC counts (*p* < 0.001) initially on day 1 post-TBI (1.2 ± 0.32 × 10^3^ cells/µL) compared to mice treated with saline (0.31 ± 0.03 × 10^3^ cells/µL). Similarly, on day 14, FSl-1-treated mice showed a significantly higher WBC count (*p* < 0.001) (0.27 ± 0.37 × 10^3^ cells/µL) compared to the saline-treated cohort (0.072 ± 0.04 × 10^3^ cells/µL).

NEUs: Mice treated with FSL-1 pre-TBI showed significantly higher neutrophils count (*p* < 0.001) (1.05 ± 0.28 × 10^3^ cells/µL on day 1 and 0.15 ± 0.02 × 10^3^ cells/µL on day 14) compared to the saline-treated cohorts (0.25 ± 0.03 × 10^3^ cells/µL on day 1 and 0.04 ± 0.03 × 10^3^ cells/µL on day 14).

LYMs: Mice treated with FSL-1 pre-TBI showed a trend of increased LYM counts on days 14 and 30 (0.072 ± 0.03 × 10^3^ cells/µL on day 14 and 2.28 ± 1.46 × 10^3^ cells/µL on day 30) compared to the saline-treated cohorts (0.025 ± 0.03 × 10^3^ cells/µL on day 14 and 0.58 ± 0.32 × 10^3^ cells/µL on day 30).

PLTs: Mice treated with FSL-1 pre-TBI showed significantly higher platelet counts (*p* < 0.001) initially on days 1, 4, and 7 post-radiation (1247.17 ± 126.3 × 10^3^ cells/µL on day 1, 1258.0 ± 184.0 × 10^3^ cells/µL on day 4 and 434.33 ± 59.87 × 10^3^ cells/µL on day 7) compared to the saline-treated cohort. The saline-treated animals showed a platelet count of 559.5 ± 177.8 × 10^3^ cells/µL on day 1, 825.83 ± 123.7 × 10^3^ cells/µL on day 4, and the lowest count of 111.5 ± 71.0 × 10^3^ cells/µL on day 7.

H&E staining of sternum bone marrow specimens from the same animal groups showed increased megakaryocyte counts for FSL-1 cohorts on days 14 and 30 post-TBI (*p* = 0.032) (Figure 2E,G). Our results show that male mice (*n* = 6 per timepoint) treated with FSL-1 pre-TBI had a megakaryocyte count of 24.4 ± 20.8 per section of sternum compared to 3.25 ± 4.03 megakaryocytes per sternum section in saline-treated mice on day 14 post-radiation. Similarly, on day 30 post-TBI, mice treated with FSL-1 showed 59.6 ± 11.5 megakaryocytes compared to 36 ± 14.14 megakaryocytes in saline-treated mice. The megakaryocyte count decreased from day 1 post-TBI, was the lowest on day 7, and recovered after day 30. The naïve animals used as a reference (dotted line) showed 137.67 ± 23.46 megakaryocytes per sternum section.

Additionally, male mice treated with FSL-1 12 h pre-TBI showed a significantly higher colony-forming capability compared to the saline cohorts, as measured by the colony-forming unit assay of femoral bone marrow (Figure 2F). Here, the total cellularity (total number of colonies per well) was used to assess the colony-forming ability of femoral bone marrow cells isolated (*n* = 6) from male mice treated with FSL-1 or saline pre-TBI exposure to mixed-field radiation (4.8 Gy). Our results show that male mice treated with FSL-1 14 days post-TBI had 88 ± 40.04 colonies per well compared to 9 ± 3.46 colonies in saline-treated mice (*p* < 0.0001 on day 14). Similarly, on day 30 post-TBI, male mice treated with FSL-1 showed 40 ± 7.8 colonies per well compared to 28 ± 5.8 colonies per well for male mice treated with saline (*p* = 0.36, not significant).

Flow cytometry analysis of femoral bone marrow cells showed elevated hematopoietic stem cell (HSC) populations in male mice treated with FSL-1 12 h pre-TBI (Figure 2H,I). Samples from six animals were pooled for this analysis. For gating the HSC populations, initially, singlet cells were gated using forward scatter (area vs. height), and lineage-negative cells were identified from this sub-population. The resulting lineage-negative cells were probed for c-kit and sca-1-positive staining to identify HSC cells [34]. Femoral bone marrow cells stained with HSC markers (c-kit+/sca-1+/lin-) showed a higher number of HSC cells in the FSL-1-treated group (556 cells per femur) compared to 240 HSC cells per femur in saline-treated mice 1 day post-TBI. Naive animals used as a reference (dotted line) showed 134 HSC cells per femur.

### 2.3. Effect of FSL-1 Administration Pre-TBI on Inflammatory Markers in Mouse Serum

To understand the effect of FSL-1 on inflammatory circulatory markers and cytokines in mice, serum separated (*n* = 6) on days 1, 4, 7, 14, and 30 post-TBI were analyzed. Serum levels of E-selectin (Figure 3A) showed a significant increase in FSL-1-treated groups on days 1 and 14, with *p*-values < 0.001 on both days. Similarly, P-selectin levels were significantly higher for FSL-1-treated mice on days 1, 14, and 30 (*p* < 0.001). Conversely, total MMP-9 showed significantly higher levels on days 1 and 3 in saline-treated cohorts compared to FSL-1-treated mice (*p* < 0.001 at both timepoints). However, this trend reversed by day 30, when FSL-1-treated mice showed higher total MMP-9 levels (*p* < 0.001).

E-selectin: The FSL-1 group showed an average of 115,893 ± 2376 pg/mL of E-selectin in serum compared to an average of 42,078 ± 7630 pg/mL in saline-treated mice on day 1 post-radiation. Our results show decreasing levels of E-selectin in serum, with the lowest levels in saline-treated mice on day 14 (5422 ± 2190 pg/mL). On the same day, E-selectin levels in the FSL-1-treated group were 14,573 ± 3587 pg/mL. E-selectin levels in the serum of naïve mice were used as a reference (60,572.67 ± 8397.316 pg/mL).

P-selectin: The FSL-1 pre-treated group showed elevated levels of P-selectin in serum on day 1 post-radiation (44,069 ± 1196 pg/mL) compared to saline-treated mice (12,418 ± 1651 pg/mL). On day 14 and day 30 post-radiation, mice treated with FSL-1 showed increased P-selectin levels in the serum (5203 ± 1152 pg/mL on day 14 and 15,189 ± 1685 pg/mL on day 30) compared to saline-treated mice (2020 ± 633.7 pg/mL on day 14 and 10,425 ± 1174 pg/mL on day 30).

FLT-3 ligand: In response to radiation exposure, FLT3 ligand (FLT-3L) levels in serum increased on days 4 and 7 in both the FSL-1 and saline-treated groups without significant difference. However, the FLT-3L levels decreased from day 14 to day 30. Saline-treated mice still showed higher levels of FLT-3L (2557 ± 226.4 pg/mL on day 14) compared to mice treated with FSL-1 (2077 ± 523.5 pg/mL on day 14), with a significance of *p* = 0.015. Naive animals with FLT-3L levels of 334.28 ± 32.24 pg/mL in the serum were used as a reference.

Total MMP-9: Mice treated with saline pre-TBI showed higher levels of total MMP-9 on day 1 compared to FSL-1-treated mice (86,887 ± 10,236 ng/mL in saline-treated mice compared to 54,209 ± 6951 ng/mL in FSL-1treated mice). Total MMP-9 decreased but still showed higher levels in saline-treated mice on day 4 (40,849 ± 10,541 ng/mL in saline-treated mice and 13,038 ± 2146 ng/mL in FSL-1-treated mice). This trend reversed on day 30, with FSL-1-treated mice showing higher total MMP-9 levels (44,319 ± 3615 ng/mL) compared to saline-treated mice (30,023 ± 6721 ng/mL). Total MMP-9 levels in naïve animals (95.768.25 ± 12,956.44 ng/mL) were used as a reference.

Erythropoietin: Erythropoietin (EPO) levels in the serum of male mice increased upon exposure to radiation. By day 14, EPO levels in serum increased to 170,075 ± 54,387 pg/mL in saline-treated mice and 73,350 ± 26,764 pg/mL in FSL-1-treated mice (*p* < 0.0001). By day 30, EPO levels in radiated animals reached closer to naïve levels (553.40 ± 103.95 pg/mL).

To assess the effect of FSL-1 on inflammatory cytokines/chemokines and growth factors, a Bioplex assay was performed with the serum samples from the same cohort (used for ELISA) using a cytokine 23-plex panel. Serum samples (*n* = 3) were used for both FSL-1 and saline groups on days 1, 4, 7, 14, and 30 post-radiation (Figure 3B). Our results show significantly increased interleukin-6 (IL-6) (*p* = 0.003), keratinocyte-derived chemokine (KC) (*p* < 0.001), and macrophage inflammatory protein 1a (MIP-1a) levels (*p* < 0.0001) in FSL-1-treated mice compared to the control group 14 days post-radiation. On the contrary, KC (*p* = 0.0003), IL-1b (*p* = 0.005), MIP-1a (*p* = 0.009), MIP-1b (*p* = 0.008), and tumor necrosis factor (TNF-a) (*p* = 0.006) showed significantly decreased levels 1 day post-radiation in mice treated with FSL-1 pre-TBI compared to mice treated with saline.

### 2.4. FSL-1 Administration Pre-TBI Prevents the Dysregulation of Toll-Like Receptor-Mediated Immune Response After Radiation Exposure

Spleens were isolated from male mice treated with FSL-1 (1.5 mg/kg) or saline 12 h pre-TBI exposure in mixed-field conditions (65% neutron and 35% gamma) at a sublethal dose of 4.8 Gy. Samples were collected (*n* = 3) on days 1, 4, 7, 14, and 30 post-radiation exposure. RNA isolated from spleens was used to synthesize cDNA, and gene expression changes were analyzed using an RT2-Profiler PCR array for a toll-like receptor signaling pathway. Samples from the irradiated saline-treated groups were labeled as RV followed by the day of collection (RV1, RV4, RV7, RV14, and RV30). Samples from the irradiated treatment (FSL-1) groups were labeled as RD followed by the day of collection (RD1, RD4, RD7, RD14, and RD30). Our results showed differential gene expression patterns in mice treated with saline or FSL-1 pre-TBI compared to naïve control mice (*n* = 3). Our results showed the highest downregulation of toll-like receptors in saline-treated male mice on day 14. Consistently downstream adaptors of toll-like receptors, like myd88 and ticam1/2, also showed downregulation in saline-treated mice on days 14 and 30 post-radiation. Similarly, genes in JAK/STAT and cytokine signaling pathways also showed high downregulation in saline-treated male mice on day 14. Overall, male mice treated with saline 12 h pre-TBI showed a higher number of downregulated genes compared to FSL-1-treated male mice (Figure 4B).

FSL-1 is a known ligand of TLR2/6 receptors [29]. We propose that FSL-1 given prophylactically to male mice exposed to mixed-field radiation activates the toll-like receptor signaling pathway and maintains the regulation of pro-inflammatory cytokines and interferon signaling through Myd88-dependent signaling, as depicted in Figure 5. In the absence of FSL-1, our results show a dysregulation of inflammatory pathways due to damage-associated patterns that activate TLR5/8 receptors intracellularly in response to radiation exposure. 

## 3. Discussion

Protecting warfighters and civilians from a nuclear or radiological accident or attack is a national security priority vital to supporting the health of military personnel and civilian populations. Advance notice of an accident or an attack is unlikely, thus underscoring the importance of treatment options that can be administered shortly after (up to days) exposure. However, warfighters and civilian first responders may have unique occupational exposures wherein they may be deployed or tasked with responding to objectives within a radiation-contaminated field. For these individuals, the approval of prophylactic countermeasures will prove invaluable. The power of nuclear weapons can be varied, and so must be the countermeasures. The detonation of a 10-kiloton nuclear device will result in a broad spectrum of damage. The neutron-to-gamma ratio for a given total dose will be dependent on the location of the detonation, weather conditions, time of day, height of the burst, and yield of the device. Dangerous levels of radiation will be released immediately and last for hours to days from the resulting fallout [35]. Based on the typical neutron-to-gamma ratio values, which in turn are based on the weapon yield and all other factors that are reported in table 2-III of the NATO Handbook [32], we have studied MF exposure at 65% neutron and 35% gamma in mice. Here, we report our findings on the efficacy of FSL-1, a synthetic lipopeptide, which may have a role as a countermeasure against lethal mixed-field radiation exposure.

In a recent publication, we demonstrated that FSL-1 given as a single subcutaneous dose 24 h pre-exposure to gamma radiation provides significant survival benefits in a murine model of total body gamma irradiation [31]. In this study, we have shown FSL-1’s efficacy in both low LET (gamma) and high LET (MF, gamma, and neutron) when administered as a single dose at 6 or 12 h prior to exposure in male and female mice. These findings show the use of FSL-1 under varying radiation exposure scenarios, such as nuclear detonation or fallout. These results, together with our previous studies with FSL-1, show a wider range of therapeutic windows (6 h to 24 h pre-exposure) as a potential prophylactic radiation countermeasure [31].

Recovery from hematopoietic damage can improve survival from radiation exposure in H-ARS. Previous studies indicate that increased platelet counts are associated with improved recovery and survival rates [36]. Platelet inducers, like romiplostim (Nplate) and thrombopoietin mimetic (TPOm), have shown efficacy as extremely potent medical countermeasures in H-ARS mouse models [33,37,38]. Our studies here show that FSL-1 given as a single dose can improve hematopoietic recovery in an H-ARS mouse model over time. Within the hematopoietic system (consisting of blood, bone marrow, and spleen), the administration of FSL-1 improved recovery in mice through elevating total WBC, NEU, LYM, and PLT counts compared to the control cohort. FSL-1 pre-TBI administration led to an increase in circulating platelet counts, which correlated with elevated megakaryocyte numbers (platelet precursors in sternal bone marrow). In femoral bone marrow, FSL-1 pre-treatment also boosted progenitor colony formation and expanded the short-term hematopoietic stem cell population. This FSL-1-induced enhancement of hematopoietic stem and progenitor cells likely contributes to its protective effect against H-ARS, potentially preventing radiation-induced immune dysfunction. In addition to damaging the hematopoietic system, H-ARS also induces changes in cytokine levels in the serum. Our results showed the impact of pre-total body irradiation (TBI) FSL-1 administration on inflammatory responses in a murine model. We observed a rapid increase in serum E-selectin and P-selectin levels one day after exposure to mixed-field radiation (neutron and gamma). This elevation aligns with E-selectin’s role in hematopoietic stem cell migration during immunosuppression and P-selectin’s involvement in platelet binding and leukocyte recruitment [39]. While selectin levels declined by day 7 post-irradiation, FSL-1 treatment facilitated their recovery by day 14. In contrast, total MMP-9 levels showed an inverse trend: an initial reduction in FSL-1-treated mice followed by an increase by day 30. The known dual role of MMP-9, mitigating H-ARS tissue damage while potentially inducing fibrosis at high levels [40], warrants further investigation. Additionally, the observed elevation in EPO in irradiated animals is indicative of radiation-induced damage. The dysregulation of inflammatory cytokines due to radiation-induced hematopoietic damage has been reported earlier. Here, FSL-1-treated animals showed elevated IL-6, MIP1a, and KC levels on day 14, which descended to normal levels by day 30. These elevated immunostimulatory cytokines may be responsible for accelerating hematopoietic recovery post-radiation exposure.

Toll-like receptors (TLRs) are established mediators of innate immune responses [29,41,42]. Ionizing radiation activates TLR signaling, and FSL-1 is a known ligand for TLR2/6 receptors [29]. The TLR signaling system is highly expressed in splenic tissue [43]. In the current study, we used spleen samples isolated from irradiated mice exposed to mixed-field radiation to study the expression of the TLR pathway-associated genes, using an RT2-Profiler PCR array. Our study here showed that the TLR signaling was activated in mice post-radiation. However, in irradiated vehicle-treated mice, a high downregulation of this signaling cascade was observed on day 14. This downregulation affected downstream components, like cytokine signaling, as well as JAK/STAT and JNK signals associated with toll-like receptor activation. This shows a clear dysregulation of the TLR system in saline-treated mice exposed to mixed-field radiation. Such dysregulation was not seen in mice treated with FSL-1 pre-TBI, suggesting that this potent countermeasure serves an immunomodulatory function. In this study, we have shown that FSL-1 protected bone marrow and radiation-sensitive tissues from radiation-induced hematopoietic injury. We plan to conduct further studies in the future to understand the effect of FSL-1 on protecting animals from radiation-induced gastrointestinal damage and reduce lethality.

## 4. Materials and Methods

### 4.1. Murine Model for Hematopoietic ARS

Pathogen-free male and female C57BL/6 mice were obtained from Jackson Laboratories (Bar Harbor, ME, USA). They were housed at the Armed Forces Radiobiology Research Institute (AFRRI) Department of Laboratory Animal Resources (DLAR) facility and were acclimated for at least 5 days prior to use in experiments. Mice received Harlan Teklad Global Rodent Diet 8604 and acidified water (pH 2.5–3.0) ad libitum and HEPA-filtered air in each individually ventilated cage, which housed up to 5 mice. Mice were housed on a 12 h light and dark cycle and were 12–14 weeks old at the start of all experiments as described earlier [44].

### 4.2. Ethics Statement

All experiments were ethically conducted following protocols approved by the Uniformed Services University of the Health Sciences DLAR Institutional Animal Care and Use Committee (IACUC) in accordance with principles outlined by the National Research Council’s Guide for the Care and Use of Laboratory Animals. The DLAR facility is accredited by the Association for the Assessment and Accreditation of Laboratory Animal Care (AAALAC) International [44].

### 4.3. Drug Preparation and Administration

Mice in the vehicle control group received 0.1 mL sterile saline ( 0.9% NaCl). Lyophilized FSL-1 (supplied by InvivoGen, San Diego, CA, USA) was reconstituted in saline and protected from light as described earlier [31]. Each mouse in the experimental group received a single dose of 1.5 mg/kg FSL-1 in 0.1 mL saline at the nape of the neck subcutaneously (SC) at the specified timepoints prior to irradiation.

### 4.4. Total Body Irradiation (TBI) Studies Using Cobalt-60

Mice were placed in restraint boxes with eight separated compartments made of Lucite dividers as described earlier [31]. Mice were irradiated bilaterally (simultaneously) at an estimated dose rate of 0.6 Gy/min in the cobalt-60 gamma-irradiation facility at the AFRRI, Bethesda, MD, USA. An alanine/Electron Spin Resonance (ESR) dosimetry system (American Society for Testing and Material Standard E 1607) was used to measure the dose rates in the cores of acrylic phantoms (3 inches long and 1 inch in diameter) located in all empty slots of the exposure rack in the Lucite restraint boxes. ESR signals were measured with a calibration curve based on standard calibration dosimeters provided by the National Institute of Standards and Technology (NIST, Gaithersburg, MD, USA). The calibration curve was verified by inter-comparison with the National Physical Laboratory (NPL) in the United Kingdom as reported previously [44].

### 4.5. Total Body Irradiation (TBI) Studies Using Mixed-Field Neutron and Gamma Radiation in the AFRRI TRIGA Facility

Mice were restrained in individual perforated aluminum tubes connected to one another forming a column. Each column of tubes rotated at ~1 rpm on the radiation platform 255 cm from the reactor. All tubes on the outer edges were filled with acrylic phantoms. Mice were irradiated unilaterally within the TRIGA irradiation facility at AFRRI, Bethesda, MD. Dosimetry was obtained from paired ion chambers calibrated in the high-level cobalt facility at AFRRI, using a standard reference chamber calibrated to the NIST (NIST, Gaithersburg, MD, USA) standard references at the Department of Medical Physics at the University of Wisconsin prior to use in the AFRRI facility. These paired ion chambers are then utilized in mapping runs to determine the neutron and gamma components of the radiation exposure in AFRRI reactor Exposure Room 1. A standard shielding configuration and associated shielding coefficients measured and reported by the NIST in the AFRRI TRIGA facility were utilized to produce the 65% neutron and 35% gamma irradiation environment [45]. All measurements were made in accordance with the American Associations of Physicists in Medicine (AAPM) TG-51, the protocol for clinical reference dosimetry of high-energy photon and electron beams, and the internal AFRRI Contract Report CR85-1, a practical guide to ionization chamber dosimetry at the AFRRI reactor [46,47]. During all mouse exposure runs, dosimetry was measured via two monitor ion chambers calibrated during paired ion chamber mapping runs to determine the total dose.

### 4.6. Housing and Care of Animals After Irradiation

Following irradiation and clearance by the AFRRI Health Physics Department to ensure TRIGA-exposed animals were no longer activated, mice were returned to their cages at the DLAR. Mice were monitored up to four times daily and scored for pain, unresponsiveness, immobility, abnormal respiration, and disheveled appearance according to criteria approved in the IACUC protocol. Mice were considered moribund and were humanely euthanized according to current American Veterinary Medical Association (AVMA) standards when they were unable to stand upright, abdominally gasped for breath, or reached a threshold sum of scores from the predetermined criteria [48]. Body weights were taken initially prior to radiation. We did not monitor body weight change throughout this study because we observed that frequent handling of animals post-radiation exacerbates their response to irradiation, so set intervals of observing body weight were not used, which is a limitation of this study. During critical periods of peak mortality, food pellets were added directly to the floor of the cage in addition to being distributed from the feeder rack. This was a means of providing ease of access to animals that might struggle to obtain the food otherwise.

### 4.7. Prophylactic Survival Efficacy with a Single Dose of FSL-1 Using Gamma and Mixed-Field Radiation in Male and Female C57BL/6 Mice

Thirty-day survival studies were conducted with two different radiation qualities-gamma only and MF radiation with 65% neutron and 35% gamma radiation for side-by-side comparison. For gamma radiation, male mice were weighed and either received a saline injection 6 h or 12 h prior to irradiation or received a single 1.5 mg/kg FSL-1 (InvivoGen, San Diego, CA, USA) injection constituted in saline 6 h or 12 h prior to irradiation. The male mice were then irradiated at 9.0 Gray (Gy) with gamma radiation. The efficacy study was confirmed in female mice with FSL-1 or its vehicle administered 12 h prior to irradiation at a dose of 8.3 Gy. For 65% neutron and 35% gamma MF radiation, mice were similarly injected once every 12 h prior to irradiation with either saline or FSL-1 at 1.5 mg/kg. Both male and female mice were irradiated at 5.5 Gy. Animals were monitored closely for clinical signs and symptoms of pain, and mice that scored past the moribund threshold as outlined earlier were humanely euthanized.

### 4.8. Harvesting Blood and Tissues for Various Molecular Assays

Male mice were given saline or 1.5 mg/kg FSL-1 12 h prior to MF irradiation at 4.8 Gy. Blood was collected from these mice via cardiac puncture from isoflurane-anesthetized mice (*n* = 6) on days 1, 4, 7, 14, and 30 after irradiation. Animals were euthanized via cervical dislocation after blood collection. Mice were dissected for sternum and femur collection. Sternums were collected from the mice (*n* = 6) on days 1, 4, 7, 30, and 60 post-irradiation for histopathological analysis of progenitor cells. On days 4, 14, 30, and 60 post-radiation, both femurs were extracted from mice (*n* = 3) for performing clonogenic assays as stated below. Spleens were harvested and snap-frozen for RT-PCR analysis as described below.

### 4.9. Hematological Recovery with FSL-1

#### 4.9.1. Analysis of Peripheral Blood Cells and Biomarkers

A small sample of EDTA-treated whole blood ~20 µL was used for complete blood count (CBC) with the HESKA Element HTTM 5 Analyzer (HESKA Corporation, Loveland, CO, USA) system, which quantified white blood cells (WBCs), neutrophils (NEUs), monocytes (MONOs), lymphocytes (LYMs), and platelets (PLTs) of each mouse. Serum was analyzed for circulatory markers of bone marrow damage by Enzyme-Linked Immunosorbent Assay (ELISA), P-selectin, E-selectin, Flt-3 ligand, MMP-9, and erythropoietin (catalog #MPS00, MES00, MFK00, MMPT90, and MEP00B) from R&D Systems (Minneapolis, MN, USA) following manufacturer’s instructions. The serum was also evaluated for inflammatory cytokines/chemokines and growth factors using the Bio-Plex Pro Assay kit to quantify 23 cytokines following the manufacturer’s (Bio Rad) instructions (Hercules, CA, USA).

#### 4.9.2. Hematopoietic Progenitor Clonogenic Assay

Bone marrow cells were flushed from femurs into centrifuge tubes with Iscove’s modified Dulbecco’s medium using 23G needles. Cells were cultured at 2 × 10^5^ cells/plate according to the manufacturer’s instructions (Mouse Colony-Forming Cell Assays Using MethoCult, 03444, Stem Cell Technologies, Cambridge, MA, USA). After 14 days of incubation, the total number of colonies was quantified using the STEMvisionTM colony counter (Stem Cell Technologies).

#### 4.9.3. Sternal Histopathology

Sterna were fixed in 10% buffered formalin for at least 24 h following organ harvest. They were next soaked in 12–18% sodium EDTA (pH 7.4–7.5) to decalcify for 3 h. Sterna were then dehydrated with graded ethanol concentrations, embedded in paraffin, and cut into 5 µm long sections. The sections were rehydrated and stained with hematoxylin and eosin (H&E) stain on microscope slides. Bone marrow was evaluated across multiple sternebrae for loss of cellularity. In addition, megakaryocytes were counted and quantified from the slides. Images were captured with an Olympus DP70 camera and imported into Zeiss ZEN (v3.8) for analysis.

#### 4.9.4. Hematopoietic Stem Cell Staining of Femoral Bone Marrow Using Flow Cytometry

Femoral bone marrow cells were harvested at different timepoints from male mice exposed to 4.8 Gy mixed-field radiation as described earlier (CFU method section). The cells from pooled samples (*n* = 6; 0.5–1 million cells per mL) were suspended in a staining buffer (00-4222-26, ThermoFisher Scientific, Waltham, MA, USA) followed by 30 min incubation at 4 °C in an antibody staining panel containing a c-kit (105827, BioLegend, San Diego, CA, USA), sca-1 (108137, BioLegend), and a lineage cocktail (78022, BioLegend). The cells were then incubated in a fix-lyse buffer (00-5333, ThermoFisher Scientific) for 10 min at room temperature followed by centrifugation at 4 °C for 5 min at 2000 rpm. The supernatant was discarded and washed with saline once and centrifuged using the same conditions again. Finally, the cells were strained using cell strainer tubes (08-771-23, Fisher Scientific) and protected from light until use. Samples from naïve animals (*n* = 6) were used as controls. For flow cytometry, an Attune NXT acoustic flow cytometer was used following the manufacturer’s instructions for instrument settings. Flow cytometry data was analyzed using FlowJo software (v10.10.0) for gating and statistical derivations. The number of hematopoietic stem cells was plotted using GraphPad Prism software.

### 4.10. RT2-Profiler PCR Array

#### RNA Extraction and cDNA Synthesis

Frozen spleen samples harvested from male mice exposed to 4.8 Gy mixed-field radiation and pre-treated with saline or FSL-1 12 h prior were used for RNA extraction. Total RNA was extracted using a mirVana miRNA isolation kit (AM1561, ThermoFisher Scientific) following the manufacturer’s instructions. The purity and yield of RNA were measured using Nanodrop One using the A260/280 method. Pure RNA was used to synthesize cDNA using the RT2 Easy First Strand Kit (330421, Qiagen, Germantown, MD, USA) following the manufacturer’s instructions. The cDNA was then mixed with RT2 SYBR Green master mix, the mixture was aliquoted on the RT2-Profiler PCR array (GeneGlobe Id: PAMM-018Z), and Quant studio 3 was used for PCR. Ct values generated for the samples were analyzed using Qiagen’s web-based GeneGlobe data analysis. The fold change was calculated from Ct values based on the 2delta-detla-Ct method. Finally, the gene regulation changes were expressed as fold regulation (log-inverse of fold change). The data was plotted as heatmaps using GraphPad Prism statistical software.

### 4.11. Statistical Analysis

Survival data was plotted as Kaplan–Meier plots. Survival experiments utilized *n* = 24–25 animals per group. For the survival data, Fisher’s exact test was used to compare survival at 30 days, and a log-rank test was used to compare survival curves with GraphPad Prism 10 software; a *p*-value of less than 0.05 was considered significant. Means and standard errors were reported for all other data. Analysis of variance (ANOVA) was used to determine if there was a significant difference among different groups.

## 5. Conclusions

There is currently an unmet need to develop radioprotectants against hematopoietic acute radiation syndrome, which can be used in scenarios with either high or low LET exposures. FSL-1 has been shown to be an effective mitigator against H-ARS in murine as well as non-human primate models exposed to total body gamma radiation [29,30,31]. Our current study showed that FSL-1 is an extremely potent radioprotectant against different radiation qualities and protects bone marrow and the immune system, thereby expanding its therapeutic window and applicability as a radiation countermeasure for H-ARS. In order to better understand the mechanism of action of FSL-1, we plan to investigate the direct effect of FSL-1 on hematopoiesis in bone marrow in the near future.

## Figures and Tables

**Figure 1 ijms-26-05303-f001:**
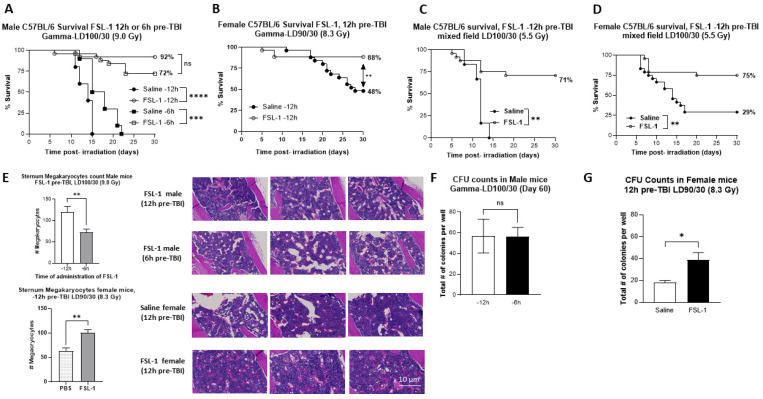
Pre-treatment with FSL-1 significantly increases survival in male and female C57BL/6 mice exposed to total body irradiation. (**A**–**D**) Kaplan–Meier survival curves in male and female mice exposed to (**A**,**B**) 9.0 Gy (male) and 8.3 Gy (female), respectively, under gamma exposure. (**C**,**D**) Mixed-field exposures at 5.5 Gy with 65% neutron and 35% gamma ((**C**,**D**) in male and female mice, respectively). Significance was tested using a log-rank p test and Fisher’s exact test, with *p* < 0.05 considered significant. (**E**) Assessment of megakaryocytes from the sternal bone marrow collected from surviving animals (from (**A**,**B**)) at the end of this study (day 60 post-TBI) with quantification of the H&E-stained sections. Representatives of these sections are presented here. (**F**,**G**) Femoral bone marrow cells from survivors (from (**A**,**B**)) were assessed on day 60 post-TBI for clonogenicity by a 14-day colony-forming unit assay. The number of colonies formed was plotted (**F**,**G**). Statistical analysis was performed using an unpaired *t*-test, and *p* < 0.05 was considered significant. * *p* ≤ 0.05, ** *p* ≤ 0.01, *** *p* ≤ 0.001, **** *p* ≤ 0.0001.

**Figure 2 ijms-26-05303-f002:**
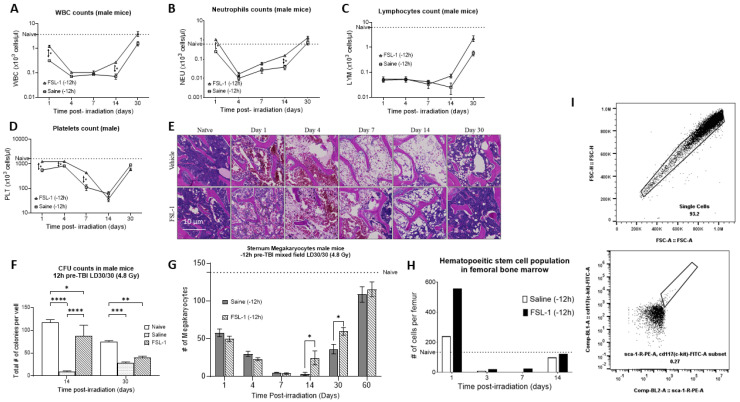
Pre-treatment with FSL-1 accelerates hematopoietic recovery in male C57BL/6 mice exposed to mixed-field radiation at 4.8 Gy. (**A**–**D**) Complete blood cell counts measured in mouse whole blood collected on days 1, 4, 7, 14, and 30 post-TBI. Statistical analysis was performed using multiple *t*-tests, and *p* < 0.05 was considered significant. (**E**) H&E-stained representative sections of the sternum collected on days 1, 4, 7, 14, and 30 post-TBI. (**F**) Colony-forming unit assay was performed using femoral bone marrow samples collected on days 14 and 30 post-TBI. Statistical significance was measured using 2-way ANOVA. (**G**) The graphical representation of megakaryocytes from the sternal sections is presented in (**E**). Multiple *t*-tests were used to establish statistical significance. (**I**) Femoral bone marrow cells were stained for KSL positive population. (**H**) The hematopoietic stem cell population in the femoral bone marrow population on 1, 4, 7, and 14 days post-TBI. * *p* ≤ 0.05, ** *p* ≤ 0.01, *** *p* ≤ 0.001, **** *p* ≤ 0.0001.

**Figure 3 ijms-26-05303-f003:**
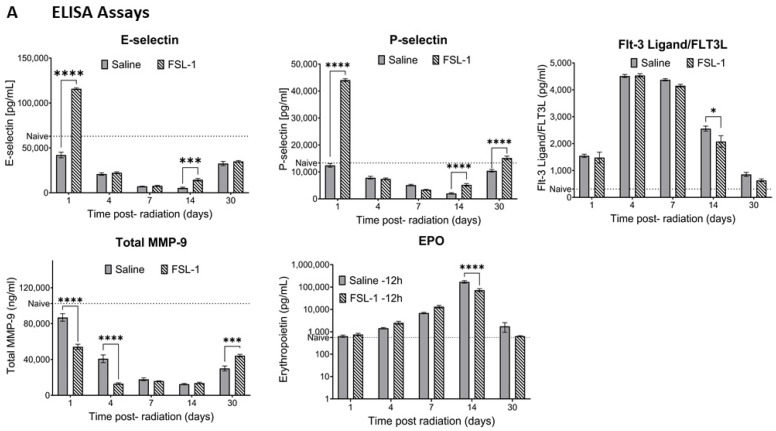
Effect of FSL-1 administration pre-TBI on inflammatory markers in mouse serum. Serum samples collected from male mice exposed to mixed-field radiation (4.8 Gy) at multiple timepoints were tested for various immune markers using ELISA assays (**A**). The same serum samples (*n* = 3) were also tested for inflammatory markers using a Bioplex assay with a cytokine 23-plex panel. Some of the markers that showed significant differences are presented here (**B**). Significance was measured using 2-way ANOVA between groups. The average of naïve samples (*n* = 3) is shown as a reference (dotted line). * *p* ≤ 0.05, ** *p* ≤ 0.01, *** *p* ≤ 0.001, **** *p* ≤ 0.0001.

**Figure 4 ijms-26-05303-f004:**
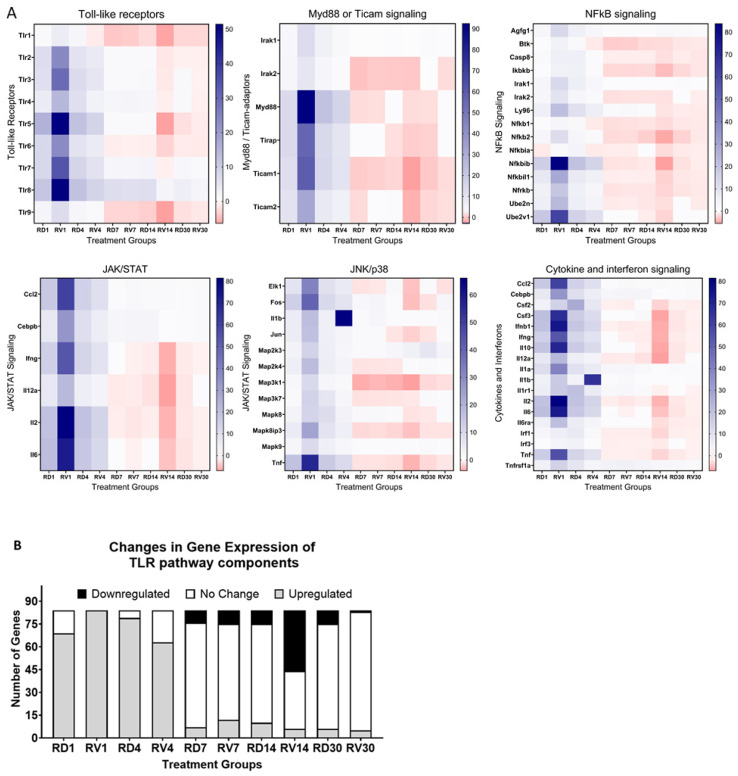
Gene expression changes over time in male C57BL/6 mice pre-treated with FSL-1 or saline 12 h before radiation. RNA extracted from spleen tissues isolated from male mice exposed to mixed-field radiation at a dose of 4.8 Gy was used for qRT-PCR using an RT2-Profiler PCR array. The Ct values were analyzed using the web-based Qiagen Gene Globe Analysis. Fold regulation was calculated from fold change based on delta–delta–Ct values. Calculated fold regulation was plotted as heatmaps using GraphPad Prism software (**A**). Based on the analysis, gene expression changes were plotted as genes upregulated, downregulated, or unchanged in sample groups (**B**). Radiated vehicle groups were labeled as RV, and radiated treatment (FSL-1) groups were labeled as RD.

**Figure 5 ijms-26-05303-f005:**
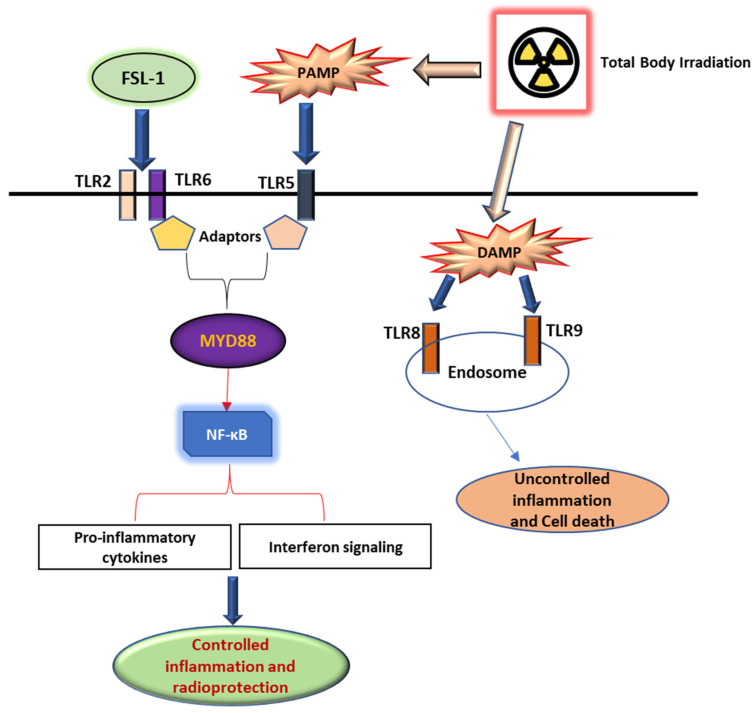
FSL-1 protects against radiation-induced damage through the TLR signaling pathway. FSL-1 given prior to radiation exposure activates the TLR2/6 receptor pathway through Myd88-dependent signaling, thereby enabling controlled pro-inflammatory cytokines and interferon signaling mediated by NF-κB. Conversely, PAMP and DAMP released extracellular and intracellular activate alternate pathways that promote the dysregulation of inflammatory signaling, resulting in cell death and fibrosis.

## Data Availability

All data generated or analyzed during this study are included in this published article.

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
