# Peer review of "FSL-1 Pre-Administration Protects Radiation-Induced Hematopoietic Organs Through the Modulation of the TLR Signaling Pathway"

_ijms, 2025, doi:10.3390/ijms26115303_

Round 1
Reviewer 1 Report
Comments and Suggestions for Authors
In this study, the authors examined the effect of FSL-1 pre-administration on hematopoietic acute radiation syndrome in mice. Pre-administration of FSL-1 prolonged survival, accelerated hematopoietic cell recovery, and modulated immune responses in mice exposed to mixed-field (MF) radiation. Although the results clearly demonstrate the preventive effect of FSL-1 against hematopoietic acute radiation syndrome, I have the following comments and questions that should be addressed by the authors.
- Figure 1. FSL1 increased survival in mice exposed to lethal gamma radiation and lethal MF radiation. It would be helpful to include peripheral blood cell count data in thee mice.
- High-dose radiation causes gastrointestinal syndrome as well as hematopoietic syndrome. Although this study focuses on hematopoietic syndrome, the authors should describe the findings of gastrointestinal damage in the mice analyzed in Figures 1 and 2. What were the food intake and body weight changes in those mice?
- Is the effect of FSL-1 on hematopoietic cells direct or indirect? The effect of FSL-1 on bone marrow hematopoietic cells should be examined in mice without radiation exposure or in cell culture experiments.
Minor
- There is no citation of Figure 2I or 2H in the main text.
- Discussion. "--- FSL-1 show a wider range of therapeutic windows" However, the authors should discuss possible adverse effect(s) of FSL-1 administration.
Author Response
Open Review (Reviewer 1)
In this study, the authors examined the effect of FSL-1 pre-administration on hematopoietic acute radiation syndrome in mice. Pre-administration of FSL-1 prolonged survival, accelerated hematopoietic cell recovery, and modulated immune responses in mice exposed to mixed-field (MF) radiation. Although the results clearly demonstrate the preventive effect of FSL-1 against hematopoietic acute radiation syndrome, I have the following comments and questions that should be addressed by the authors.
- Figure 1. FSL1 increased survival in mice exposed to lethal gamma radiation and lethal MF radiation. It would be helpful to include peripheral blood cell count data in these mice.
Response: Thank you for your comment. We have observed previously that serial blood collection from animals in survival studies impacts their health by exacerbating their response to radiation sickness, which may lead to early mortality. Therefore, blood samples were taken at the end of the study period. This data was not included as control vehicle group in both male studies did not have survivors at the completion of study.
- High-dose radiation causes gastrointestinal syndrome as well as hematopoietic syndrome. Although this study focuses on hematopoietic syndrome, the authors should describe the findings of gastrointestinal damage in the mice analyzed in Figures 1 and 2. What were the food intake and body weight changes in those mice?
Response: As the primary objective of these studies (Figure 1) was hematopoietic ARS and recovery in presence/absence of countermeasure, we did not collect gastrointestinal (GI) tissues to test GI damage. Studies in Figure 2 were done at a much lower dose (~LD30) to compare both vehicle and FSL-1 groups through day 30 to ensure survival in both groups. Based on our previous studies, peak lethality due to GI-ARS occurs from days 4-10. Most of the animals in the studies reported here were found dead or euthanized around 15-20 days range, which is the hematopoietic death range in this strain of mice (PMID: 35241733). We have an established a partial body irradiation (PBI) model (for C57BL/6 mice) that we intend to use for evaluating the efficacy of FSL-1 in GI-ARS in near future. (PMID: 38014611 DOI: 10.1667/RADE-23-00068.1).
Body weights were taken initially prior to radiation. We have observed that frequent handling of animals, post-radiation exacerbates their response to irradiation so set intervals of observing body weight were not used.
During critical periods of peak mortality, food pellets were added directly to the floor of the cage in addition to being distributed from the feeder rack. This was a means of providing ease of access to animals that might struggle to obtain the food otherwise.
- Is the effect of FSL-1 on hematopoietic cells direct or indirect? The effect of FSL-1 on bone marrow hematopoietic cells should be examined in mice without radiation exposure or in cell culture experiments.
Response: Thanks to the Reviewer for the question. We do not know the direct effect of FSL-1 on hematopoiesis in bone marrow since we did not have a group with the drug alone without radiation. In near future, we plan to apply for Intramural/Extramural funding to investigate the direct mechanism of action of FSL-1 on hematopoiesis.
Minor
- There is no citation of Figure 2I or 2H in the main text.
Citation for Figure 2H-I has been added in the methods section. (https://doi.org/10.1002/cyto.a.20674).
- Discussion. "--- FSL-1 show a wider range of therapeutic windows" However, the authors should discuss possible adverse effect(s) of FSL-1 administration.
No adverse effects of FSL-1 has been observed in our previous studies. Safety evaluation of FSL-1 in non-radiated mice was previously investigated and published by our group (PMID: 38373011 DOI: 10.1667/RADE-23-00142.1.).

Reviewer 2 Report
Comments and Suggestions for Authors
Title: FSL-1 pre-administration protects radiation-induced hematopoietic organs through modulation of TLR signaling pathway
In this study, the author demonstrated that FSL-1 significantly enhances survival and promotes hematopoietic recovery in mice exposed to mixed neutron and gamma radiation, underscoring its potential as a preventive treatment. The experimental design and the presentation of the results effectively support the conclusions. However, I have a few comments before accepting for publication.
- The author should provide the catalog number for Mouse Colony-Forming Cell Assays Using MethoCult; Stem Cell Technologies; Cambridge, MA.
- The author should add details on how the megakaryocytes from the sternal bone marrow collected from a surviving animal were assessed.
- The author should add details on the hematopoietic stem cell population in the femoral bone marrow population.
- KSL gating is correct in fig. 2I. So the author should check this.
- Why does the author use femoral bone marrow in this study? Why not include tibiae?
Author Response
Open Review (Reviewer 2)
Title: FSL-1 pre-administration protects radiation-induced hematopoietic organs through modulation of TLR signaling pathway
In this study, the author demonstrated that FSL-1 significantly enhances survival and promotes hematopoietic recovery in mice exposed to mixed neutron and gamma radiation, underscoring its potential as a preventive treatment. The experimental design and the presentation of the results effectively support the conclusions. However, I have a few comments before accepting for publication.
- The author should provide the catalog number for Mouse Colony-Forming Cell Assays Using MethoCult; Stem Cell Technologies; Cambridge, MA.
Response: The catalog number for the methocult product used was 03444 (Stem Cell technologies; Cambridge, MA). This has been added to the Methods section.
- The author should add details on how the megakaryocytes from the sternal bone marrow collected from a surviving animal were assessed.
Response: Megakaryocytes from the images of H&E stained sternum bone marrow sections were counted manually using Zeiss ZEN (v3.8). The counted megakaryocytes per group were averaged and statistical analysis was done using graphpad prism software.
- The author should add details on the hematopoietic stem cell population in the femoral bone marrow population.
Response: The details were added to the methods section
- KSL gating is correct in fig. 2I. So the author should check this.
Response: The KSL cells were identified after gating out lineage negative cells. We have used c-kit/sca-1 dual positive cells out of lineage-negative populations to identify KSL positive cells.
- Why does the author use femoral bone marrow in this study? Why not include tibiae?
Response: Although both femurs and tibia have been used to study hematopoietic stem cell populations, our preference to use femoral bone marrow in this study was due to dual use of femurs in our downstream applications. Same cell populations from femoral bone marrow were used for both 14-day bone marrow colony formation assay (2F-2G) as well as KSL staining.

Round 2
Reviewer 1 Report
Comments and Suggestions for Authors
The authors have provided appropriate responses in the response letter. However, these points should be included in the main text, either discussed in the Discussion section or stated as limitations.
Minor
Line 204. Is (2H-I) (Figure 2H & I)?
Author Response
2nd Round : Open Review (Reviewer 1)
Comments and Suggestions for Authors
The authors have provided appropriate responses in the response letter. However, these points should be included in the main text, either discussed in the Discussion section or stated as limitations.
Answer: Thanks to the Reviewer for the suggestion, we truly appreciated it. We included the points in the manuscripts as green highlights in the Discussion, Conclusion, and Methods section.
Minor
Line 204. Is (2H-I) (Figure 2H & I)?
Answer: Corrected (please see green highlight)
